# A computational toolbox for the assembly yield of complex and heterogeneous structures

Agnese I. Curatolo[1], Ofer Kimchi [2], Carl P. Goodrich[3], Ryan K. Krueger [1] & Michael P. Brenner [1,4] ✉

The self-assembly of complex structures from a set of non-identical building blocks is a hallmark of soft matter and biological systems, including protein complexes, colloidal clusters, and DNA-based assemblies. Predicting the dependence of the equilibrium assembly yield on the concentrations and interaction energies of building blocks is highly challenging, owing to the difficulty of computing the entropic contributions to the free energy of the many structures that compete with the ground state configuration. While these calculations yield well known results for spherically symmetric building blocks, they do not hold when the building blocks have internal rotational degrees of freedom. Here we present an approach for solving this problem that works with arbitrary building blocks, including proteins with known structure and complex colloidal building blocks. Our algorithm combines classical statistical mechanics with recently developed computational tools for automatic differentiation. Automatic differentiation allows efficient evaluation of equilibrium averages over configurations that would otherwise be intractable. We demonstrate the validity of our framework by comparison to molecular dynamics simulations of simple examples, and apply it to calculate the yield curves for known protein complexes and for the assembly of colloidal shells.

The hierarchical assembly of complex building blocks underpins much of biology, allowing the spontaneous formation of protein complexes, virus shells, and structural components of the cell with high accuracy and without external influence. A hallmark of these assembly processes is that they are often *heterogeneous*: individual components making up the assembled product are different from each other, having different shapes and binding characteristics. Such heterogeneous components store information about their assembly processes via their highly tuned interactions. Heterogeneous self-assembly contrasts with classical homogeneous self-assembly, whereby large numbers of identical components interact to form materials.

In recent years, it has become possible to synthesize heterogeneous components in the laboratory. Examples range from nanostructures either coated with[1] or entirely composed of DNA[2,3], to proteins with rationally designed binding interfaces[4,5], to lithographically printed blocks with magnetic interactions[6], to colloids coated with hydrophilic and hydrophobic patterns[7,8], to shape complementarity of colloids with tunable depletion forces[9,10].

The potential design space of heterogeneous self-assembly is enormous. If the interactions between individual entities can be chosen at will, then determining the choices leading to a desired assembly target (or more generally an interesting emergent behavior) can be

[1]School of Engineering and Applied Sciences, Harvard University, Cambridge, MA 02138, USA. [2]Lewis-Sigler Institute, Princeton University, Princeton, NJ 08544, USA. [3]Institute of Science and Technology Austria, A-3400 Klosterneuburg, Austria. [4]Department of Physics, Harvard University, Cambridge, MA 02138, USA. ✉e-mail: brenner@seas.harvard.edu

quite difficult. A popular approach has been to choose individual components such that the desired structure is the minimal free energy state of the system[11]. In a variety of systems, from DNA to proteins, this approach has led to the assembly of complex structures. However, even when components are chosen so that the desired structure is the ground state, crosstalking interactions cause non-desired structures to form[12,13] or incomplete structures can be more favorable under certain conditions. Given specific interactions, there is a fundamental limit on the size of complex structures that can reliably assemble out of heterogeneous components[14,15]. With increasing numbers of components, there is a yield catastrophe above which the yield of the desired structure decays exponentially. For polymeric structures, the yield catastrophe can be controlled by tuning the individual concentrations of the different components[15], but finding assembly strategies that achieve high yield for non-polymeric systems remains difficult.

Robust strategies for solving these problems in the laboratory are unknown. Biology shows that robust solutions exist, but has had millions of years of evolution to find the best designs. For synthetic systems, the time and monetary costs of experimentally testing various self-assembly mixtures to determine those with highest yields can be significant.

A typical route towards addressing these challenges is to use molecular simulations. Simulations could allow rapid screening of the design space of heterogeneous assembly in particular experimental systems, including how the shape, binding characteristics and relative concentrations of the building blocks contribute to the desired emergent property[16]. Significant advances in computational software have made it possible to simulate assemblies of components with nontrivial shapes and interactions. However, such simulations, especially for large heterogeneous structures, are often prohibitively expensive. The use of simulations for exploring heterogeneous assembly has therefore been limited.

Another approach to predict the equilibrium self-assembly yield is to calculate it analytically. While for spherical particles with isotropic interactions the partition functions for small clusters can be calculated analytically[17], no such analytical calculation exists for anisotropic interactions. We hypothesized that automatic differentiation could be leveraged to perform this otherwise intractable calculation[18–20]. In automatic differentiation, the execution of a computer program is accompanied by the construction of a computation graph of primitive operations whose derivatives are known and can therefore be recombined (via the chain rule) to compute the gradient of the larger program. This procedure can be applied recursively, allowing us to efficiently evaluate higher-order derivatives of nearly any computer function with machine accuracy.

The goal of this paper is to develop a combined analytical/computational approach for calculating the concentration- and temperature-dependencies of equilibrium assembly yield for heterogeneous building blocks with complex geometries. We show that this approach enables us to calculate the relevant entropic factors—vibrational, rotational and translational—and estimate the equilibrium assembly yield of the structures.

This paper is organized as follows. First, we present the core statistical mechanical model, extending prior theories for the self-assembly of identical spherical colloids to heterogeneous components with complex geometries. We show that several terms that cannot be calculated purely analytically can be accurately calculated by leveraging newly-developed computational automatic differentiation tools (JAX)[21,22]. Next, we compare the model to simulations to explore its regime of validity and sources of potential error. We apply our procedure (depicted in Fig. 1) to two biological protein assemblies, the PFL and TRAP complexes, and predict the temperature- and concentration-dependence of their equilibrium assembly. Finally we demonstrate that our method can be applied to compute the yield of large multimeric structures such as cages.

Basic algorithms and code are open-sourced at Github[23].

## Results

### The analytical model

**Defining the yield.** We consider a cluster comprised of $N_s$ building blocks with short range interactions. Each building block $i$ has three translational degrees of freedom $(q_{ix}, q_{iy}, q_{iz}) = \vec{q}_i$ and three rotational ones, represented by the three Euler angles $(\varphi_i, \theta_i, \psi_i) = \vec{\phi}_i$. The potential energy of the cluster $E_s(\{\vec{q}, \vec{\phi}\})$ is thus a function of $6N_s$ coordinates.

The equilibrium properties of the cluster are determined by its partition function

$$Z_s = \frac{1}{\sigma_s} \int_{\Omega_s} \left( \prod_{i=1}^{N_s} d^3\vec{q}_i d^3\vec{\phi}_i \right) e^{-\beta E_s\left(\left\{\vec{q}, \vec{\phi}\right\}\right)} \tag{1}$$

where $\Omega_s$ is the region of phase space where the cluster is defined, and the symmetry number $\sigma_s$ accounts for all possible combinations of rotations and building block permutations that result in the same cluster $s$[24]. Although (1) is a well-known result[17] we also derived it more fully in "Methods—Deriving the equilibrium yields" for completeness.

(1) describes the properties of a single cluster. In a self-assembly process, however, many clusters coexist and compete with each other to recruit building blocks. The question we ask is: at equilibrium, if we pick a cluster at random, what is the probability that we pick cluster $s$? The observable quantity associated with such probability is the equilibrium yield $Y_s$. In experiments or agent-based numerical simulations where we can enumerate the number $n_s$ of clusters of type $s$, the yield can be simply defined as

$$Y_s = \frac{n_s}{\sum_{s'} n_{s'}}, \tag{2}$$

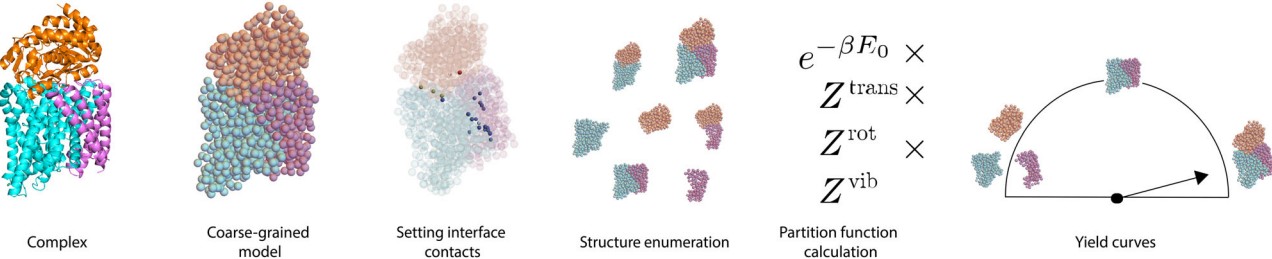

**Fig. 1 | Overview of the analysis procedure.** We depict the process by which we predict assembly yield as a functions of system parameters, using the TRAP protein complex as an example. Starting with a given complex (here, a PDB file), we generate a coarse-grained model (here, each amino acid is replaced by a sphere). We specify the contacts at the various binding interfaces and their strengths (here, using patches at the interfaces). We then enumerate all possible structures that can form (in this case, 3 monomers, 3 dimers, 1 trimer). Finally, we compute the partition function for each structure as described in this work, and compute the expected yields of each structure as a function of system parameters. The true yield curves for the TRAP protein complex are shown in Fig. 4.

Below the figure, labels: Complex | Coarse-grained model | Setting interface contacts | Structure enumeration | Partition function calculation | Yield curves

In the partition function panel: $e^{-\beta F_0} \times$, $Z^{\text{trans}} \times$, $Z^{\text{rot}} \times$, $Z^{\text{vib}}$

where the sum in the denominator runs over all possible occurring structures $s'$.

Within the grand canonical ensemble, the yield can be expressed in terms of the normalized concentrations $\tilde{c}_\alpha$ of the different building block species, proportional to their fugacities[25] (see "Methods–Deriving the equilibrium yields" for further details):

$$Y_s = \frac{\left(\prod_\alpha \tilde{c}_\alpha^{N_{s,\alpha}}\right) Z_s}{\mathcal{Q}} \equiv \frac{\mathcal{Q}_s}{\mathcal{Q}}, \tag{3}$$

where $N_{s,\alpha}$ is the number of building blocks of type $\alpha$ in structure $s$. The grand partition function $\mathcal{Q}$ of the system is then the sum of $\mathcal{Q}_s$ over all clusters that the system can form[25].

**The partition function calculation.** For simple models, e.g., if the building blocks have rotational symmetry and the attractive energy is isotropic, the integral in (1) can be directly computed numerically. For more complicated problems, that involve e.g., anisotropic potentials and non-trivial rotational degrees of freedom, a possible route is to compute the integral by sampling (e.g., with Monte Carlo techniques). This is computationally expensive, with a cost that grows with the number of substructures and the complexity of components. Another possibility is to *analytically* compute these terms. While historically such calculations have been highly challenging, we show that using modern automatic differentiation tools makes this straightforward. The analytical approach is more efficient, and also gives insight into the physics of the different terms in the partition function.

We assume that the building blocks can have arbitrary shape, and that their interactions can be localized to specific regions (interfaces) of the building block. We restrict our calculation to only treat rigid clusters, where we use the term "rigid" as opposed to "floppy" to indicate a cluster without zero modes (i.e., internal degrees of freedom about which movement incurs no energetic cost)[26]. Accounting for such internal degrees of freedom requires a challenging and nontrivial calculation of the relevant entropic factor beyond the scope of this current work. A geometric formulation of the entropic factor resulting from such floppy modes has been addressed for the case of isotropic potentials with short interaction range in ref. 27. Intuitively, the free energy of a system with $n$ zero modes is represented as an $n$-dimensional manifold whose boundaries can be determined from the collection of configurations with $(n-1)$ zero modes.

We start our analytical calculation by performing a change of coordinates in (1) that allows us to express the integral in terms of center of mass (COM) coordinates of the cluster. These new coordinates are 3 global translations $\vec{q}_c$ and 3 global rotations $\vec{\phi}_c$ of the cluster, and $6N_s - 6$ internal vibrations $\xi_i$, as a consequence of the rigid cluster assumption. These new coordinates automatically take into account the effective phase space $\Omega_s$ over which the integral in $Z_s$ needs to be performed, so that it does not need to be specified explicitly. The change of coordinates reads

$$Z_s = \frac{1}{\sigma_s} \int d^3\vec{q}_c \int d^3\vec{\phi}_c \int d^{6N_s-6}\vec{\xi}\, J\left(\vec{q}_c, \vec{\phi}_c, \vec{\xi}\right) e^{-\beta E\left(\vec{\xi}\right)} \tag{4}$$

where $J$ is the Jacobian of the coordinate transformation and $E$ the energy of the cluster–where we have dropped the subscript $s$. In this reference system, we assume that the potential energy of the cluster does not depend on the global translations and rotations, but only on the internal vibrations. In order to perform the integral, we need to compute $E(\vec{\xi})$ and $J(\vec{q}_c, \vec{\phi}_c, \vec{\xi})$.

The Jacobian introduced in (4) is the matrix of partial derivatives of a function $f : \mathbb{R}^{6N} \to \mathbb{R}^{6N}$ such that $f(\nu) = \mu$, where $\mu = \{\vec{q}, \vec{\phi}\}$ are the coordinates in the building blocks' reference frame and $\nu = \{\vec{q}_c, \vec{\phi}_c, \vec{\xi}\}$ are the coordinates in the cluster's reference frame. In order to define $f$ we first compute the eigenvalues and eigenvectors of

the Hessian of the energy. We then define a function $\tilde{f} : \mathbb{R}^{6N-6} \to \mathbb{R}^{6N}$, which corresponds to the transpose of the matrix of eigenvectors associated with the nonzero eigenvalues. Thus, $\tilde{f}(\tilde{\nu}) = \tilde{\mu}$, where $\tilde{\nu} = \vec{\tilde{\xi}}$ and $\tilde{\mu}$ represent the internal degrees of freedom in the building blocks reference frame. Finally, we apply a rotation $R(\vec{\phi}_c)$ and a translation $T(\vec{q}_c)$ to $\tilde{f}$ to yield the function of the variables transformation $f$:

$$f = T \circ R \circ \tilde{f}. \tag{5}$$

We note that the Jacobian of $f$ turns out to be a simple function of the moment of inertia only if the individual building blocks do not have rotational degrees of freedom, e.g., in the case of uniformly DNA-coated colloids[17]. However, if the building blocks have rotational degrees of freedom that depend on the global rotations of the cluster (in the case of proteins that can only bind with specific orientations), then the Jacobian assumes a more complicated form (see "Methods–The Jacobian of a dimer with rotational degrees of freedom").

*Translational entropy.* To start with, we consider the Jacobian's dependence on the translational coordinates $\vec{q}_c$. By definition, the COM positions are a linear transformation of the building blocks' coordinates ($\vec{q}_c = \sum_i \vec{q}_i / N$), which implies that the Jacobian is independent of the global translations. We can thus readily perform the integral over the translations $\vec{q}_c$ which simply yields the system volume $V$, so that we obtain

$$Z_s = \frac{V}{\sigma_s} \int d^3\vec{\phi}_c \int d^{6N_s-6}\vec{\xi}\, J\left(\vec{\phi}_c, \vec{\xi}\right) e^{-\beta E\left(\vec{\xi}\right)}. \tag{6}$$

*Vibrational entropy.* Secondly, we look at the integrand's dependence on the vibrational modes $\vec{\xi}$. We consider the case in which the thermal energy is much smaller than the potential energy of the cluster. In this case, the vibrations are small and the integral is dominated by the minimum of the potential energy $E_0$ of the cluster. We can then apply Laplace's approximation[28], which yields

$$Z_s = e^{-\beta E_0} \frac{V}{\sigma_s} \left(\prod_{i=1}^{6N_s-6} \sqrt{\frac{2\pi}{\beta \omega_i^2}}\right) \int d^3\vec{\phi}_c\, J\left(\vec{\phi}_c\right), \tag{7}$$

where the $\omega_i^2$'s are the eigenvalues of the Hessian of the energy $E(\vec{\xi})$, calculated via automatic differentiation using the function `jax.hessian()` at $\vec{\xi} = \vec{0}$ (see "Methods–Computing the partition function with automatic differentiation"). Note that $J(\vec{\phi}_c)$ is also evaluated at $\vec{\xi} = \vec{0}$, which constrains the building blocks' relative positions and orientations to those that define the cluster $s$.

Here, we have taken Laplace's approximation to lowest (second) order. The error arising from this approximation will increase for less parabolic energy landscapes such as can occur for more esoteric interaction potentials, more complex interfaces, or even for longer-range interactions (Supplementary Fig. 3). As energy minima become less parabolic, higher-order corrections need also be considered. Such corrections can be computed using the inverse of the Hessian matrix alongside higher-order derivatives[28]. In the Numerical Results section we explore and comment on the validity of Laplace's approximation in this context.

*Rotational entropy.* The remaining integral, of the Jacobian over the global rotations, is performed numerically. First, we uniformly sample $10^5$ values of $\vec{\phi}_c$ using the quaternion representation, to avoid problems that arise from sampling Euler angles, such as non-uniform distribution of orientations, singularities, and the gimbal lock problem[29]. It may be possible to achieve high accuracy with fewer calculations but we found that $10^5$ samplings were sufficient to yield accurate analytic predictions. For the results presented in this manuscript, this sampling procedure took < 30 seconds of compute time and < 3 MB of memory (as measured by the `tracemalloc` library) on a personal laptop computer (e.g., 26s and 2 MB for the TRAP complex

trimer presented subsequently in the manuscript). We then convert the sampled quaternions to Euler angles—using the $\{x, y, z\}$ (fixed) convention—so that rotations are represented with only three variables. Although it might in principle be possible to express the Jacobian of the change of rotational variables analytically (see "Methods—The Jacobian of a dimer with rotational degrees of freedom"), this is highly non-trivial in 3D, and we instead use automatic differentiation[21,22] (see "Methods—Computing the partition function with automatic differentiation"). This approach allows us to numerically evaluate the Jacobian at each value of the rotations previously sampled. We define the result as

$$\tilde{J} = \int d^3 \vec{\phi}_c J(\vec{\phi}_c) \qquad (8)$$

and obtain our final expression for $Z_s$:

$$
\begin{aligned}
Z_s &= e^{-\beta E_0} \times V \times \frac{\tilde{J}}{\sigma_s} \times \prod_{i=1}^{6N_s-6} \sqrt{\frac{2\pi}{\beta \omega_i^2}} \\
&\equiv e^{-\beta E_0} \times Z_s^{\text{trans}} \times Z_s^{\text{rot}} \times Z_s^{\text{vib}},
\end{aligned}
\qquad (9)
$$

where we have manually separated the results of the integrals into *translational*, *rotational* and *vibrational* partition functions.

**Finding a self-consistent solution for the yields.** Once we have calculated the partition functions of the individual structures, we aim to calculate the structures' yields using (3). For systems with low self-assembly yields, an accurate estimate of the yield can be found by treating the concentrations in (3) as the total individual monomer concentrations. However, as yields increase, the reservoir of monomers is depleted, such that the concentrations are no longer well-approximated by the total monomer concentrations added to the system.

We therefore seek a self-consistent solution for the yields of the different structures, while imposing conservation laws for each of the monomer species. There is one conservation law for each monomer species, given by

$$\sum_s N_{s,\alpha} c_s = c_\alpha^{\text{tot}} \qquad (10)$$

where $c_s$ is the concentration of structure $s$, $c_\alpha^{\text{tot}}$ is the total concentration of monomer $\alpha$, and as previously $N_{s,\alpha}$ is the number of monomers of type $\alpha$ in each structure $s$. Note that the unbound monomer is a valid equilibrium structure and is therefore included in the above sum.

To supplement the conservation laws, we require a linearly-independent equation for each non-monomeric structure whose yield we seek to estimate. These equations are most easily formulated as $V c_s = \mathcal{Q}_s$. Using (3), this equality can be rewritten as a mass action equation[30–32]:

$$\frac{V c_s}{\prod_\alpha (V c_\alpha)^{N_{s,\alpha}}} = \frac{Z_s}{\prod_\alpha Z_\alpha^{N_{s,\alpha}}}. \qquad (11)$$

where $c_\alpha$ is the concentration of unbound monomer $\alpha$, and $Z_\alpha$ is the partition function of the monomer (see "Methods—Deriving the equilibrium yields" for a derivation). Note that $N_s = \sum_\alpha N_{s,\alpha}$ is the total number of building blocks constituting structure $s$. The factors of $V$ cancel out with corresponding factors in the partition functions, such that the concentration predictions are independent of system volume.

We find self-consistent solutions to these equations using the `fsolve` package in `scipy.optimize`. The yield can then be calculated directly from (2) using the concentrations.

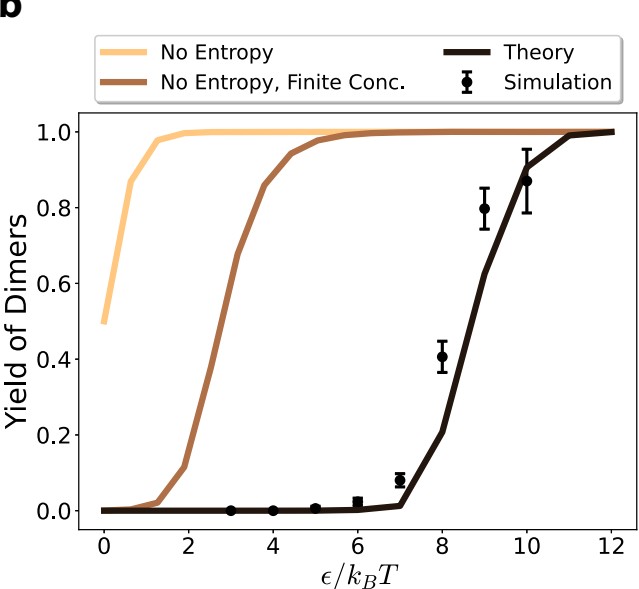

**Fig. 2 | Toy model for self-assembly of non-spherically symmetric building blocks. a** The two monomer types are shown; each colored patch of the first monomer is attracted by the corresponding colored patch of the second monomer by a Morse potential (see "Methods—Details of the pair potential"). The total attractive potential of the cluster is given by $E_0 = 3E_b$ where $E_b$ is the strength of each patch. **b** Comparison between theoretical and simulation yield for the dimeric state (when the two monomers are attached to each other). The number of building blocks of each type is $N_1 = N_2 = 9$ and the volume of the system is $V = 18,000 d^3$ where $d = 1$ is the diameter of the gray spheres. The interaction range was set to $8/\alpha = 8d/5$. Error bars represent standard error over $n = 10$ simulations, where the error is measured relative to the mean. The theoretical yield shown is computed in the canonical ensemble (see Supplemental section E). All simulations were performed in the HOOMD-blue simulation package[33].

## Numerical results

**Comparison to molecular dynamics simulations.** We perform numerical simulations to test the validity of (9). For simplicity, we consider a system of building blocks that can only assemble to form dimers in a unique way (see Fig. 2a). This allows us to non-ambiguously define a monomeric and a dimeric state and assign one of them to each structure we observe in the simulation, without having to worry about multimeric formation.

We choose a geometry for the building blocks that ensures rigidity of the dimeric state. Each monomer is composed of three spheres rigidly bound to each other, which softly repel other monomers' spheres (see "Methods—Details of the pair potential"). Each sphere is attached to a small colored patch (orange, cyan and purple). There are two types of monomers that differ by how the patches are arranged (left- and right-handed). A left-handed monomer binds to a right-handed monomer with similarly-colored patches attracting (see Methods). The presence of the three patches ensures the absence of zero modes; if each building block had only one patch, a zero mode would appear corresponding to the rotation of the monomers around the axis connecting their centers of mass.

In order to compare the analytical theory to simulation most directly, we operate in the canonical ensemble, where we can perform

exact calculations corresponding to finite sized simulations (see "Methods−The yield in the canonical ensemble"). The comparison between simulations and theory is shown in Fig. 2b. We also show the predictions arising from neglecting the contributions of entropy to the partition function, with $Z_d/Z_m = e^{3\beta E_b}$. The result of $Y_d = Z_d/(Z_d + Z_m)$ is shown in brown, while a prediction accounting for finite concentrations using Eqs. (10) and (11) is shown in orange. All simulations were performed in the HOOMD-blue simulation package[33].

We observe that the theory, when accounting for entropy, largely agrees with the simulations, but there are small quantitative discrepancies. Since the canonical ensemble calculates the yield exactly, these deviations must stem from the computation of the configurational partition function itself. Investigation reveals that the error arises from Laplace's approximation (7) which was based on the assumption of small thermal energy compared to the cluster potential energy (see Methods and Supplementary Fig. 2). This error decreases with smaller interaction ranges which are often experimentally tunable, for example by changing ionic conditions for protein self-assembly (Supplementary Fig. 3). This error can be almost completely resolved by taking Laplace's approximation to 4th order following ref. 28. In what follows we do not implement calculations to this order, but we emphasize that with the automatic differentiation tools, this is an algorithmically and computationally efficient extension of the present work.

**Predicting protein complex yield.** Our procedure for computing yields can be applied to any heterogeneous self-assembled system, provided that certain conceptual steps depicted in Fig. 1 are followed. To illustrate this, we apply these methods to compute the interaction energy- and concentration-dependent yields of protein complexes. As a starting point, we need the structure of the protein complex as given by a PDB file, and also a model of the energy of interactions between contacting residues. To illustrate the approach, here we use a model of interactions between protein complexes that is based on sequence covariance information[34], assigning a score to each pair of amino acids corresponding to the probability and strength of contact. However we emphasize that our algorithm could be used with any model of protein interactions to predict yield curves, for example Rosetta[35], AlphaFold[36] or other methods[37].

We begin by using the PDB file to define coarse-grained, rigid building blocks: each amino acid in the protein is represented by a sphere whose position is a non-weighted average of the positions of the amino acid's atoms. Similarly to Fig. 2a, the interface contacts are defined by patches placed on the interface (see Fig. 1), subjected to an attractive pair potential (see "Methods−Details of the pair potential"). For example, if it is known that amino acid $a$ belonging to protein A forms a bond with amino acid $b$ belonging to protein B and their positions in the complex are $(x_a, y_a, z_a)$ and $(x_b, y_b, z_b)$ respectively, then a patch on each protein building block will be set at position $(x_a + x_b, y_a + y_b, z_a + z_b)/2$; these two patches specifically attract one another. The interaction range is set at $8/\alpha = 8d/2$ where $d$ is the diameter of each sphere representing an amino acid. Since the patches are defined so as to minimize energy, no additional minimization procedure (e.g., simulation) is required to define the ground state.

Here we consider two protein complexes with interactions that were characterized by ref. 34: the Pyruvate formate lyase-activating enzyme complex (PFL) and the tripartite ATP-independent periplasmic (TRAP) transporter. For each predicted contact $b$ between building blocks $X$ and $Y$, we place a patch with an attractive potential $E_{XY}^{(b)}$ which we assume is proportional to $p_{XY}^{(b)}$, the probability of the contact $b$ found in ref. 34. $E_{XY}^{(b)}$ is therefore given by $E_{XY}^{(b)} = \epsilon p_{XY}^{(b)}$ where $\epsilon$ is a proportionality constant with units of $k_B T$. Since we do not know a priori what this proportionality constant $\epsilon$ should be, we treat $\epsilon$ as a parameter we vary.

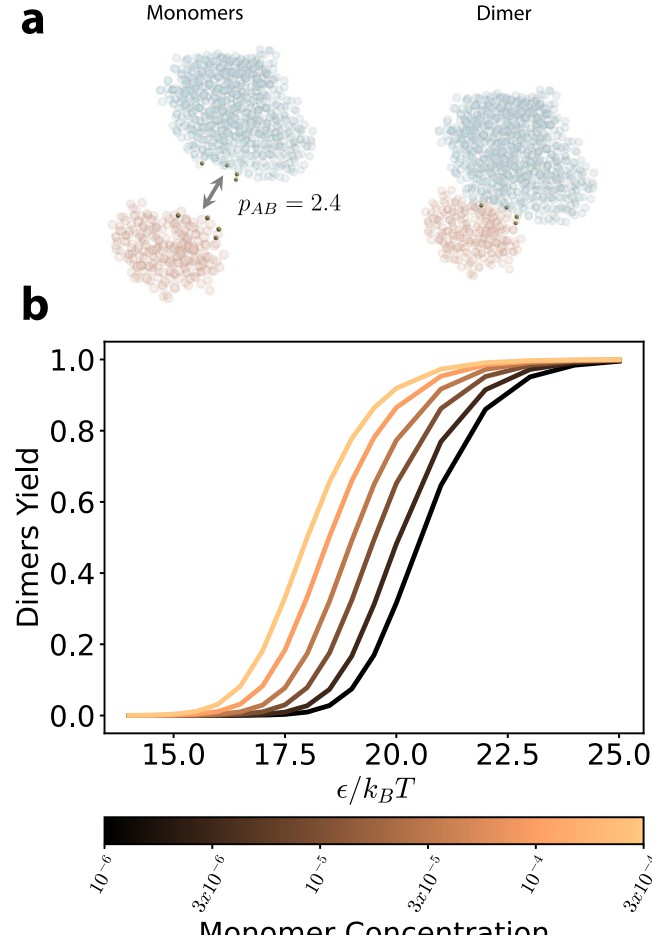

**Fig. 3 | Model and results for the PFL complex. a** Coarse grained model for proteins A (orange) and B (cyan) of the PFL complex. Left and right depict the monomeric and the dimeric complexes, respectively. The residues highlighted at the interface are the patches we put as contacts. The total strength of the AB interface is $E_{AB} = \epsilon\, p_{AB}$ where $p_{AB} = \sum_b p_{AB}^{(b)} = 2.4$ in the units of ref. 34 and $\epsilon$ is a proportionality constant that converts between these units and $k_B T$. **b** Plot showing the yield curves of the dimeric state as a function of the energy. Different colors correspond to different concentrations, here expressed in units of $d^{-3}$ where $d$ is the diameter of each sphere representing an amino acid. Source data are provided as a Source data file.

We first consider the PFL complex, composed of building blocks $A$ and $B$ bound as a dimer. For this complex, following the findings in ref. 34, we place 4 patches on each building block with relative strengths determined by the sequence covariance (see Fig. 3a). We compute the partition function for the monomeric and dimeric states using (9), and thereby compute the yield in the grand-canonical ensemble as a function of $\epsilon$ and the monomer concentrations. For simplicity, we consider equal concentrations of both monomers. The resulting yield curve is shown in Fig. 3: as expected, the dimeric complex has a higher yield at higher concentrations. For example, for $\epsilon \approx 20\, k_B T$ (corresponding to energies of individual contacts ranging from $E_{AB}^{(b)} = 6\, k_B T$ to $E_{AB}^{(b)} = 20\, k_B T$) the model predicts self-assembly (i.e., dimer yield >50%) for concentrations $\gtrsim 10^{-5} d^{-3}$ (-16.6 mM).

We next apply our formalism to the TRAP complex, a trimer comprised of three proteins referred to as M, N, and O. In order to reduce the possibility of zero modes, we place 20 patches on each interface; these patches and their strengths are chosen as the top 20 predicted contacts at each interface and their probabilities found by ref. 34. We compute the partition functions of all possible structures: the three monomers, the three dimers (MN, NO and MO), and the

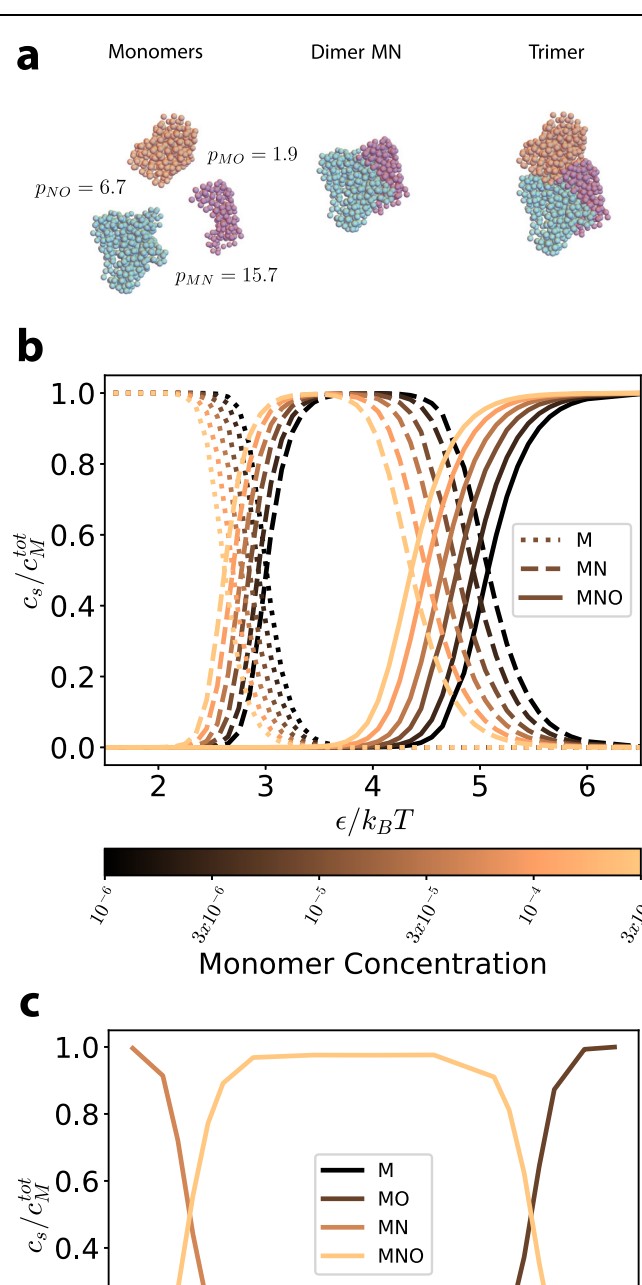

**Fig. 4 | Model and results for the TRAP complex. a** Coarse grained model for proteins M (purple), N (cyan) and O (orange) of the TRAP complex. Left, center and right depict the monomeric, the MN dimeric and the trimeric complexes, respectively. The total strengths of the three interfaces are $p_{MN} = 15.7$, $p_{MO} = 1.9$ and $p_{NO} = 6.7$. **b** We show the concentrations of the three structures which include the building block M as a function of the interaction energy. The concentrations are normalized by the total input concentration of M, $c_M^{tot}$, which was set equal to the input concentration of the other two monomers. Different colors correspond to different total concentrations $c_M^{tot}$ (in units of $d^{-3}$). When the temperature is appropriately tuned, we observe an intermediate state between the formation of only monomers and only trimers, corresponding to high yield of the dimer MN. **c** The relative concentrations of monomers M, dimers MN and MO and trimer MNO are reported for different values of $D_X$ (see main text) with fixed $\epsilon = 4\,k_BT$ and total concentration of monomers $c_M^{tot} = 10^{-5}\,d^{-3}$. Source data are provided as a Source data file.

trimer (MNO). We denote the total energy of the complex as $E_{tot} = E_{MO} + E_{NO} + E_{MN}$ where $E_{XY} = \sum_b E_{XY}^{(b)}$ is the total energy of the interface between building blocks $X$ and $Y$. From our calculation, we observe that there are three possible regimes (Fig. 4a, b): at weak interaction energies ($E_{tot} \lesssim 50\,k_BT$, corresponding to $\epsilon \lesssim 2\,k_BT$) the monomeric state is prevalent; at strong energies ($E_{tot} \gtrsim 130\,k_BT$; $\epsilon \gtrsim 5\,k_BT$) the trimeric state is prevalent; and at intermediate energies ($E_{tot} \sim 85\,k_BT$; $\epsilon \sim 3.5\,k_BT$) dimers of type MN form with high yield, while O remains as a monomer. In terms of the yields, this means that at weak energies, the monomers each form with yield 1/3; at intermediate energies the MN dimer and O monomer each form with yield 1/2; and at strong energies the MNO trimer forms with yield 1 (Supplementary Fig. 4).

We now consider the scenario where the interface contacts (hence the interface energies) can be modified, for example through mutagenesis[38–40]. We apply our model to a modified version of the TRAP complex where we let the relative energy of the MN and MO interfaces vary while keeping the overall energy $E_{tot}$ constant. We define $D_X$ to be a measure of the relative strength between these interfaces such that $E_{tot} = D_X E_{MO} + E_{NO} + D_Z E_{MN}$ where $D_Z$ is defined such that $E_{tot}$ is kept constant for each independently varied $D_X$. A plot of the relative concentrations as a function of $D_X$ are shown in Fig. 4c, where we have used $\epsilon = 4\,k_BT$ and a concentration of $c = 10^{-5}\,d^{-3}$ for all monomers. The point $D_X = 1$ corresponds therefore to the point $\epsilon = 4\,k_BT$ in Fig. 4b, where the dimers MN and the monomer O have the highest yield. As we increase $D_X$ and concurrently decrease $D_Z$, we first observe the appearance of the trimeric state, and then, at $D_X \sim 7$, an additional transition towards a state where the dimers MO and the monomer N are prevalent. This result illustrates the predictive power of our method, that allows us to compute the yield of a complex where the energy of the interfaces, and even those of the individual contacts, can be modified at will.

**The yield of spherical cages.** Finally, we show that our algorithm can be used to efficiently compute the yield of more complicated complex-forming systems, such as cages. The robust and predictable self-assembly of proteins or molecules into cages or shells plays a major role in many medical applications, from multivalent antigen presentation[41–43] to gene delivery vectors[44]. A meaningful yield landscape can be obtained provided than the enumeration step of our method is performed carefully, by taking into account the physics and biochemistry of the system under study.

We present a notional example of such an assembly, in an effort to demonstrate how this formalism works. We consider a cage of radius $R = 1$ that is assembled out of 60 spherical building blocks with icosahedral symmetry. We use a set of previously reported coordinates to define the ground state of each assembly[45]. All spheres are identical and interact via a smoothed Morse potential (see "Methods—The Jacobian of a dimer with rotational degrees of freedom") which is isotropic, so that they do not experience any rotational constraint. We set the attractive potential between spheres to a constant $E_b = \epsilon$. The minimum and cutoff for each pair of spheres are set at their nearest neighbor distance $r_0 = 0.46R$, and $r_{cut} = 0.75R$, respectively. Because the spheres interact with an isotropic potential, it was not necessary to use (8) to calculate the rotational partition function, and we instead calculate this rotational component from the moments of inertia[17]. We consider an instance of each $N$-mer (from monomer to 60-mer, see Fig. 5a) where each $(N + 1)$-mer is created by adding one building block to the $N$-mer. Such building block is chosen at random, with the only constraint that it forms at least 2 bonds with the spheres already present in the $N$-mer, to maximize the cluster's rigidity.

We predict the yield curves of these $N$-mers for different concentrations (in units of $R^{-3}$) and interaction energies. We observe a smooth transition from a monomeric state to a 60-meric state as the concentration and the interaction energy increase (see Fig. 5b), whereas the yield of the intermediate states (from dimer to 59-mer)

 

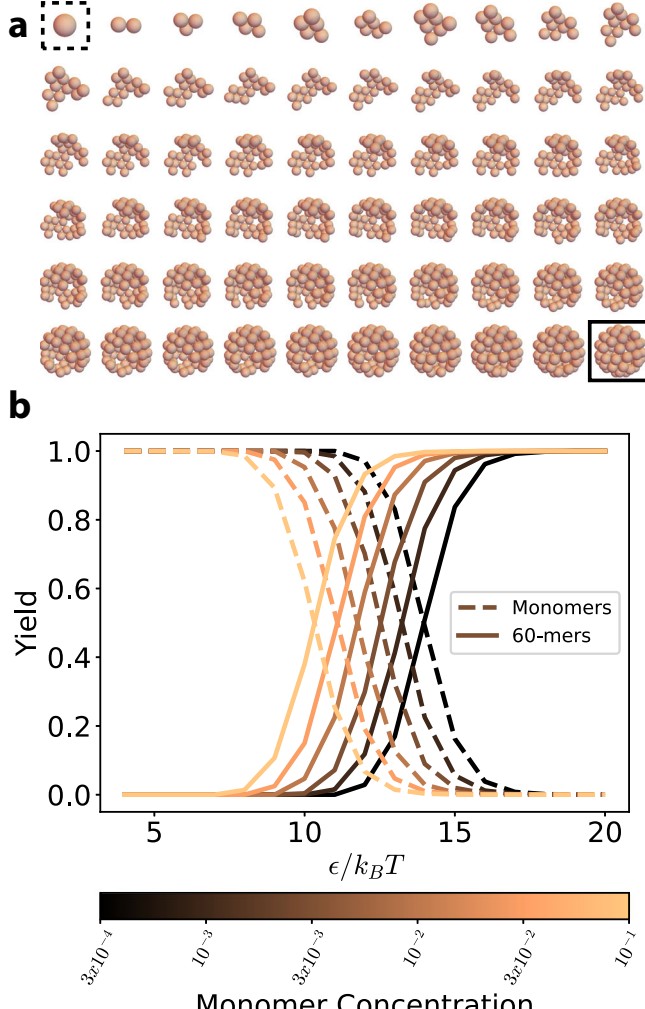

**b**

**Fig. 5 | The yield of spherical cages. a** An instance of each *N*-mer, from monomer (dashed square) to 60-mer (full square). Each pair of spheres interacts via a smoothed Morse potential with parameters $\alpha = 5R^{-1}$, $r_0 = 0.46R$, $r_{cut} = 0.75R$, where *R* is the radius of the full cage. For each intermediate, the total energy is equal to the sum of the pair potentials of all pairs of spheres, multiplied by the prefactor $\epsilon$. **b** Yield curves for monomers and 60-mers at different concentrations (in units of $R^{-3}$). No intermediate state was prevalent within the observed parameters range. Source data are provided as a Source data file.

always stays low, with a highest value—across all intermediates, all concentrations and all energies—of $8.2 \times 10^{-4}$.

## Discussion

Many structural and functional properties of biological systems rely on heterogeneous self assembly. Increasingly, building blocks with highly tuned geometries and interactions can be made experimentally, presenting an opportunity to design complex self-assembling systems with properties of living matter[46–48]. However, for increasing numbers of individual components, the yield of the desired product is diluted by an exponential number of off-target assemblies. Even if each of the off-target assemblies forms with a lower probability than the desired product, the multiplicity of these off-target assemblies can cause a yield catastrophe in which the desired product forms with negligible yield[49]. An efficient method for predicting assembly yield would enable practitioners to design self-assembling systems of increasing complexity while mitigating this yield catastrophe.

In this work, we have developed a combined analytical/computational approach to calculate the equilibrium assembly yields of complexes comprised of heterogeneous building blocks with arbitrary geometries. Our approach involves two novel calculations: (i) computing the partition function of each (predefined) candidate complex (see Eq. (9)) and (ii) given these partition functions, calculating equilibrium assembly yields (see Eqs. (10) and (11)). The classical statistical mechanics methodology we put forth can be realized by modern automatic differentiation techniques enabled by advances in machine learning[21]. While previously, the efficient calculation of the entropic parts of the partition function has been intractable except for simple cases, these advances make this calculation possible even for structures with complicated building blocks or with a large number of different components.

Our theoretical framework provides significant conceptual and practical improvements over simulation-based methods, the only other general-purpose method for yield prediction. Calculations via our approach require a drastically reduced computational cost compared to simulations (i.e., seconds or minutes vs. hours, weeks, or months). Moreover, simulation-based methods can be fraught with additional difficulties such as issues in simulating finite concentrations or efficiently sampling from equilibrium distributions[50]. Lastly, since our method involves the direct calculation of the partition function, our approach can also be used to compute other statistical properties of an equilibrium thermodynamic system (e.g., heat capacity, energy fluctuations) without complicated modifications to the calculation. Indeed, since the gradient of a solution to Eqs. (10) and (11) can be computed implicitly[51], our method could enable inverse design with respect to related thermodynamic equilibrium properties.

We have demonstrated the validity of this model's approximations by comparison to molecular dynamics simulations, where we found that precise agreement between the simulations and the analytical calculations requires paying close attention to higher order corrections to the Laplace approximations. We have also applied our methodology to two illustrative protein complexes (PFL and TRAP), as well as a simple example of a cage-forming system. We emphasize that our method can be applied to arbitrary proteins complexes so long as we know how to construct the building blocks, calculate the energies of contacts, and enumerate the most likely structures. The latter is in principle a difficult problem: enumerating all possible combinations of building blocks can become infeasible for complexes comprised of large numbers of building blocks.

We next plan to apply this method to self-assembling systems that are the subject of active study, such as de novo designed proteins and viral capsids[52]. In fact, when designing de novo protein complexes, significant care must be taken to mitigate the self-assembly of off-target structures[49]. To address this problem in silico, our model can be combined with state-of-the-art protein structure prediction and a suite of existing energy functions for describing protein-protein interactions[53–55].

Finally, as we have described, we have restricted our calculations to equilibrium systems. However, biological systems frequently make use of non-equilibrium control elements, including the regulation of the production machinery for the assembled components, as well as other control elements that are not yet available in synthetic systems such as allosteric interactions. These control knobs provide more ways of regulating energetic and entropic interactions leading to self assembly. Future work may seek to provide an understanding of how these control elements contribute to high self-assembly yields, and thus test which would be most effective to prioritize developing in a synthetic context.

## Methods
### Deriving the equilibrium yields
We consider a system composed of *N* building blocks with short range interactions. Each building block *i* has three translational degrees of freedom $(q_{ix}, q_{iy}, q_{iz}) = \vec{q}_i$ and three rotational ones, represented by

the three Euler angles $(\varphi_i, \theta_i, \psi_i) = \vec{\phi}_i$. The potential energy of the system $U(\{\vec{x}\})$, where $\vec{x}_i = (\vec{q}_i, \vec{\phi}_i)$, is then a function of $6N$ degrees of freedom. The overdamped Langevin equation that describes the dynamics of these $6N$ variables reads

$$\dot{\vec{x}} = -\frac{1}{\gamma} \vec{\nabla} U\left(\left\{\vec{x}\right\}\right) + \sqrt{2D}\, \vec{\eta}(t). \tag{12}$$

Here, $\gamma$ is the friction coefficient; $D = (\beta\gamma)^{-1}$ is the diffusion coefficient; $\beta = 1/k_B T$ where $k_B$ is the Boltzmann constant and $T$ is the temperature of the system; and $\eta(t)$ is a $6N$-dimensional Gaussian noise vector such that $\langle \eta_i(t)\eta_j(t') \rangle = \delta_{ij}\delta(t - t')$. The corresponding Fokker-Planck equation for the probability $P(\{\vec{x}\}, t)$ reads

$$\dot{P} = \frac{1}{\gamma} \vec{\nabla} \left[ \left( \vec{\nabla} U \right) P \right] + D \vec{\nabla}^2 P, \tag{13}$$

where we have dropped the $\vec{x}$ and $t$ dependencies to lighten the notation. The steady state probability $P^\star$ of the system existing in a particular phase space configuration, found by setting $\dot{P} = 0$, is

$$P^\star\left(\left\{\vec{x}\right\}\right) = \mathcal{Z}^{-1} e^{-\beta U\left(\left\{\vec{x}\right\}\right)} \tag{14}$$

where $\mathcal{Z}^{-1}$ is a normalization factor.

The total probability of $N_s \leq N$ building blocks all belonging to the same cluster $s$ is given by the integral of $P^\star$ over a phase volume $\Omega_s$, corresponding to the region of phase space where the cluster is defined. This volume includes all fluctuations that keep the identity of the cluster unchanged. We write this total probability as $\mathcal{Z}_s/\mathcal{Z}$, where $\mathcal{Z}_s$ is the *configurational* partition function of the cluster $s$, defined as

$$\mathcal{Z}_s = \frac{1}{\sigma_s} \int_{\Omega_s} \left( \prod_{i=1}^{N_s} \frac{d^3 \vec{q}_i}{\lambda_i^3} d^3 \vec{\phi}_i \right) e^{-\beta E_s(\{\vec{q}, \vec{\phi}\})}. \tag{15}$$

where $E_s$ is the potential that applies to cluster $s$. Equation (15) introduces the symmetry number $\sigma_s$, which accounts for all possible combinations of rotations and building block permutations that result in the same cluster $s$[24]. We also introduce the normalization parameter $\lambda_i$, which has units of length, so that the configurational partition function $\mathcal{Z}_s$ is unitless. The values of the $\lambda_i$'s cancel out of any equation describing the classical physical observables considered here[56].

## The grand canonical yield
Here, we are interested in finding an analytical definition of the yield of a cluster based solely on its structural details, which are encoded in its partition function $\mathcal{Z}_s$. In particular, the yield is proportional to the partition function: $Y_s \propto \mathcal{Z}_s$. To compute the proportionality factor, we need to make a choice of how to describe the system of building blocks. In other words, we need to define an ensemble.

A natural choice is the grand canonical ensemble. In the grand canonical ensemble, the number of building blocks $n$ in the system is not fixed, but the concentrations of each monomer species are. The grand partition function $\mathcal{Q}$ of the system is then the sum over all microstates with a given number of building blocks[25]. In our case, a microstate corresponds to a cluster $s$ with $N_s$ building blocks:

$$\mathcal{Q} = \sum_s \mathcal{Q}_s$$
$$\mathcal{Q}_s = \left( \prod_\alpha e^{\beta\mu_\alpha N_{s,\alpha}} \right) \mathcal{Z}_s, \tag{16}$$

where $\mu_\alpha$ is the chemical potential of building blocks of species $\alpha$ and $N_{s,\alpha}$ is the number of $\alpha$ building blocks in cluster $s$.

In our grand canonical definition, the yield $Y_s$ of cluster formed by $N_s$ building blocks can then be defined as

$$Y_s = \frac{\left( \prod_\alpha e^{\beta\mu_\alpha N_{s,\alpha}} \right) \mathcal{Z}_s}{\mathcal{Q}}. \tag{17}$$

A practical way to compute the yield defined in (17) is to express it in terms of the concentrations $c_\alpha$ of the different building block species rather than their chemical potentials $\mu_\alpha$. For point particles, these are related by $c_\alpha = e^{\beta\mu_\alpha}/\lambda_\alpha^3$[25].

For non-point particles, monomers generally have free energy $\Delta G_\alpha$. In the systems we consider here, this free energy arises from rotational degrees of freedom, such that $e^{-\beta\Delta G_\alpha} = \int d^3\vec{\phi}_i 1$. More generally, such a free energy can also arise from multiple conformations allowed to a protein, from internal secondary structure of an RNA molecule, from energetic sources such as interactions between a monomer and solvent, or from other factors. For such a monomer, its chemical potential is related to its concentration by $c_\alpha = e^{\beta\mu_\alpha} e^{-\beta\Delta G_\alpha}/\lambda_\alpha^3$. We can therefore define a normalized concentration $\tilde{c}_\alpha = c_\alpha e^{\beta\Delta G_\alpha}$, so that $\tilde{c}_\alpha = e^{\beta\mu_\alpha}/\lambda_\alpha^3$. We use this definition to rewrite (17) as

$$Y_s = \frac{\left( \prod_\alpha \tilde{c}_\alpha^{N_{s,\alpha}} \right) Z_s}{\mathcal{Q}}, \tag{18}$$

where

$$Z_s = \frac{1}{\sigma_s} \int_{\Omega_s} \left( \prod_{i=1}^{N_s} d^3\vec{q}_i d^3\vec{\phi}_i \right) e^{-\beta E_s\left(\left\{\vec{q}, \vec{\phi}\right\}\right)} \tag{19}$$

is the *unnormalized* configurational partition function (without the normalizing factors of $\lambda$). This equation is identical to (1). With this equation, we can recognize another way to equate the normalized monomer concentrations $\tilde{c}_\alpha$ to the true concentration $c_\alpha$:

$$\tilde{c}_\alpha = c_\alpha e^{\beta\Delta G_\alpha} = c_\alpha \frac{V}{Z_\alpha}, \tag{20}$$

where $Z_\alpha$ is the partition function of the monomer defined using (1).

## Deriving the self-consistent equations to solve
While the monomer conservation laws (10) are perhaps self-evident, (11) may benefit from further discussion. A useful starting point is that the ratio of the equilibrium concentrations of two species is equivalent to the ratio of their yields. We can consider one of these species to be a non-monomeric structure $s$, and the other to be a monomeric structure $\alpha$. Using the definition of yield (3) and of normalized concentrations (20), we can write this equality as

$$\begin{aligned}
\frac{c_s}{c_\alpha} &= \frac{Y_s}{Y_\alpha} \\
&= \frac{\mathcal{Q}_s}{\mathcal{Q}_\alpha} \\
&= \frac{Z_s \prod_{\alpha'} \tilde{c}_{\alpha'}^{N_{s,\alpha}}}{Z_\alpha \tilde{c}_\alpha} \\
&= \frac{Z_s \prod_{\alpha'} \left( \frac{V}{Z_{\alpha'}} c_{\alpha'} \right)^{N_{s,\alpha}}}{Z_\alpha \frac{V}{Z_\alpha} c_\alpha}.
\end{aligned} \tag{21}$$

This equation leads us directly to the equality $n_s \equiv V c_s = \mathcal{Q}_s$. Furthermore, by moving all partition functions to one side of the

equality and all concentrations to the other, we arrive at

$$\frac{c_s}{\prod_\alpha c_\alpha^{N_{s,\alpha}}} = \frac{Z_s/V}{\prod_\alpha (Z_\alpha/V)^{N_{s,\alpha}}}. \tag{22}$$

which is equivalent to (11).

An alternative derivation can be constructed following a free energy based analysis used in ref. 57 (specifically Eqs. 16 and 17 of that work). The normalized free energy of the system can be written as

$$\beta F = V \sum_s \left[ \beta c_s f_s + c_s \log\left(\frac{V c_s}{e}\right) \right] \tag{23}$$

where $f_s$ is the free energy of structure $s$, and the last term is the mixing entropy. In equilibrium, the system finds a free energy minimum subject to conservation laws for each monomeric species $\alpha$, which can be solved for using Lagrange multipliers, which we label as $\beta V \mu_\alpha$:

$$0 = \frac{\partial \left[ \beta F - \beta V \sum_\alpha \mu_\alpha \left( \sum_{s'} c_{s'} N_{s',\alpha} - c_\alpha^{tot} \right) \right]}{\partial c_s}$$
$$= \beta V f_s + V \log(V c_s) - \beta V \sum_\alpha \mu_\alpha N_{s,\alpha}. \tag{24}$$

Solving for $c_s$ yields

$$V c_s = e^{-\beta f_s} \prod_\alpha e^{\beta \mu_\alpha N_{s,\alpha}}. \tag{25}$$

To solve for the Lagrange multipliers $\mu_\alpha$, we consider the monomeric structures $\alpha$, for which (25) simplifies to

$$e^{\beta \mu_\alpha} = \frac{V c_\alpha}{e^{-\beta f_\alpha}}. \tag{26}$$

Plugging back into (25), we arrive at

$$V c_s = e^{-\beta f_s} \prod_\alpha \left( \frac{V c_\alpha}{e^{-\beta f_\alpha}} \right)^{N_{s,\alpha}}, \tag{27}$$

which, after substituting partition functions for free energies, is equivalent to (11).

## The Jacobian of a dimer with rotational degrees of freedom

Let us consider a 2D building block formed by a sphere of radius $a$ with two patches at position $(a\cos(\phi), a\sin(\phi))$ and $(a\cos(\phi), -a\sin(\phi))$, corresponding to the orange and cyan patches, respectively, shown in Supplementary Fig. 1. Patches of different colors interact with an attractive potential $U(r) = k/2r^2$, where $r$ is the distance between the patches. Each monomer can be described by two Cartesian coordinates and one angle (orientation), so that a dimer can be described by the coordinates of the two monomers that form it: $(\vec{r}_1, \vec{r}_2) = (x_1, y_1, \theta_1; x_2, y_2, \theta_2)$. In the cluster's center of mass reference frame, the dimer is described by $v = (x_c, y_c, \theta, q_1, q_2, q_3)$, where $(x_c, y_c)$ are the global translations, $\theta$ is the global rotation and $(q_1, q_2, q_3)$ are the three internal vibrations. We construct $f$ as described in (5) and calculate the Jacobian at $q_i = 0$:

$$J = 2\sqrt{2}\, a \sqrt{\cos(\phi)^2 + 1}. \tag{28}$$

We note that for the same exact system but without the patches and without the rotational degrees of freedom, the Jacobian is $J = 2\sqrt{2}\, a \cos(\phi) = 4\sqrt{I}$ where $I$ is the moment of inertia $I = \sum_i |\vec{r}_i - \vec{r}_c|^2$.

## Computing the partition function with automatic differentiation

**The Hessian.** Calculating the Hessian of $E(\vec{\xi})$ is a standard practice and it can be done by hand, if the energy function is simple enough, or numerically, for instance using finite difference methods. In the first case, for each and every system under study, the Hessian will have to be recalculated to account for the system's specific potential; in the second case, numerical differentiation introduces truncation and roundoff errors that cannot be eliminated. The automatic differentiation tools in JAX can compute derivatives of any order of any function with machine accuracy and without the need to change the code if a different potential energy function is used[58].

**The Jacobian.** The function $f$ introduced in the main text Eq. (5) is highly nontrivial for 3D structures composed of an arbitrary number of components. However, calculating its partial derivatives to obtain the Jacobian of the transformation (Eq. (4) in the main text) is fairly easy thanks to the function `jax.jacfwd(f)`, evaluated at a given set of $(\vec{q}_c, \vec{\phi}_c, \vec{\xi})$.

## Details of the pair potential

For the systems considered, the individual monomers interact via a pair potential. The attractive part of the pair potential, governing the interactions of same-colored patches on different building blocks, is:

$$E(r) = E_b (e^{-2\alpha r} - 2e^{-\alpha r}) S(r),$$

$$\text{with } S(r) = \begin{cases} 1 & \text{if } r < r_{on} \\ \frac{(r_{cut}^2 - r^2)^2 (r_{cut}^2 + 2r^2 - 3r_{on}^2)}{(r_{cut}^2 - r_{on}^2)^3} & \text{if } r_{on} < r < r_{cut} \\ 0 & \text{if } r > r_{cut} \end{cases} \tag{29}$$

where $r_{on}$ was set to 0 throughout, and the interaction range $r_{cut}$ to $8/\alpha$ unless specified otherwise. The minimum free energy of each pair potential is thus $-E_b$, at a separation of $r = 0$ between the patches. For the cage system, we use a more general form of (29) where we substitute $r$ with $r - r_0$, with $r_0$ the distance of the nearest neighboring spheres.

The monomers also include soft repulsive interactions to avoid overlaps. These interactions govern the body of the monomers (i.e., not the patches) and are given by:

$$H(r) = \frac{A}{2.5d}(d - r)^{2.5} \tag{30}$$

where we use $A = 500$ (in units of [energy]/[distance]$^{1.5}$) throughout and $d$ is the diameter of the spheres that constitute the building blocks in our coarse-grained models. $\alpha$ was set to $2/d$ or $5/d$ for different systems as described in the main text.

## The yield in the canonical ensemble

The grand canonical ensemble is natural to large protein systems; however, it introduces errors for finite systems. In order to compare our theory to finite-sized molecular dynamics simulation, we use the canonical ensemble. In this ensemble, we specify the numbers $n_i$ of each building block type $i$ in our system. These building blocks can bind to one another to form a configuration of clusters $\{c\}$, containing $N_a$ clusters of type $a$, each containing $N_{a,i}$ building blocks of type $i$. The sets of clusters that can form are therefore restricted such that

$$\sum_{a \in \{c\}} N_a N_{a,i} = n_i. \tag{31}$$

The *unnormalized* probability weight $p^{\{c\}}$ of a given set of clusters $\{c\}$ forming reads

$$p^{\{c\}} = M^{\{c\}} \prod_{N_a \in \{c\}} \left(\mathcal{Z}_a\right)^{N_a} \qquad (32)$$

where $M^{\{c\}}$ is a combinatorial coefficient that describes the number of ways of permuting the individual building blocks to arrive at $\{c\}$, and $\mathcal{Z}_a$ is the partition function introduced in (15).

The *normalized* probability – which can be read as the yield – of configuration $\{c\}$ is then

$$P^{\{c\}} = \frac{p^{\{c\}}}{\sum_{\{c\}'} p^{\{c\}'}} \qquad (33)$$

where $\{c\}'$ are all possible configurations given a fixed set of building blocks. (31) ensures that the $\lambda$ terms in $\mathcal{Z}_a$ cancel with one another in this equation.

The yield of a specific cluster $s$ in the canonical ensemble is then defined as

$$Y_s^C = \sum_{\{c\}} \frac{N_s^{\{c\}}}{\sum_{s'} N_{s'}^{\{c\}}} P^{\{c\}} \qquad (34)$$

where $N_s^{\{c\}}$ represents the number of clusters of type $s$ in configuration $\{c\}$.

For a system comprised of only dimers and monomers, the configuration $\{c\}$ is uniquely determined by the number of dimers in the system. If there are a total of $N_b$ black building blocks and $N_w$ white building blocks, which dimerize to form $N_d$ dimers, $M^{\{c\}}$ is given by

$$M_{\text{dimer system}}^{\{c\}} = \binom{N_b}{N_d}\binom{N_w}{N_d} N_d! \qquad (35)$$

**Comparison for exactly solvable system**

In order to test whether Laplace's approximation is at the root of the discrepancy between theory and simulations, we need to be able to calculate $Z_s$ exactly, that is, without the help of Laplace's approximation. We therefore consider a simpler system shown in Supplementary Fig. 2A, with clusters that are rotationally invariant: a spherical black monomer is attracted to a spherical white monomer, and they form a spherical gray dimer when their centers of mass coincide, with no excluded volume effect. The configurational partition function of such a dimer is a simple integral over a sphere.

In Supplementary Fig. 2 we compare the yield calculated with the exact $Z_s$ and the one calculated with Laplace's approximation (detailed derivation to follow). While the former shows excellent agreement with the simulation, Laplace's approximation introduces errors qualitatively and quantitatively similar to those observed in the non-rotationally-invariant system (Fig. 2). Furthermore, we verify that extending Laplace's approximation to a higher order (4th order) gives a result in between the two. Thus, we confirm that Laplace's approximation is responsible for a systematic discrepancy with simulations.

**The exact and approximated calculations for spherical dimers**

In this sub-section we describe the calculation portrayed in Supplementary Fig. 2 in more detail. The system in question consists of $N_1$ black spheres and $N_2$ white spheres. Spheres of like color repel, while spheres of opposite color attract, such that in addition to the two monomer species, a dimer species consisting of overlapping black and white spheres can form.

Because of the simplicity of the system, $Z_s$ can be exactly calculated for both the monomer species and the dimer. We start from (1), where for the dimer species $E_s$ describes the attraction between a black

and white sphere (for the monomers, $E_s$ is zero). As described in (29), $E_s$ is a function only of the distance between the centers of the two spheres; the two spheres in the dimer have the same rotational freedom enjoyed by monomers. Therefore, the contribution of rotational degrees of freedom to the partition function will cancel out in equations representing physical observables (34) much like how factors of $\lambda$ cancel out. Our quantity of interest is therefore

$$\frac{Z_{\text{dimer}}}{Z_1 Z_2} = \frac{\int d^3\vec{q_1} d^3\vec{q_2} e^{-\beta U_{\text{dimer}}\left(\vec{q_1}-\vec{q_2}\right)}}{V^2} \qquad (36)$$

where $Z_1$ and $Z_2$ are the partition functions of the two monomers.

By changing to COM coordinates ($\vec{q_i} = \vec{q_1} - \vec{q_2}$ and $\vec{q}_{\text{COM}} = \vec{q_1} + \vec{q_2}/2$), and rewriting our system in spherical coordinates, we find the exact solution

$$\frac{Z_{\text{dimer}}}{Z_1 Z_2} = \frac{4\pi \int r^2 e^{-\beta U(r)} dr}{V} \qquad (37)$$

where $U(r)$ is the attractive potential rewritten in spherical coordinates as in (29).

Our ultimate goal, however, is to determine the error introduced by Laplace's approximation. We now therefore rewrite the system following the protocol outlined in the main text. Instead of changing to spherical coordinates, we expand $U_{\text{dimer}}(\vec{q_i})$ to second order. Letting $q_i = |\vec{q_i}|$,

$$U_{\text{dimer}}(\vec{q_i}) = -E_0 + E_0 \alpha^2 q_i^2 + \mathcal{O}(q_i^3). \qquad (38)$$

Here, $E_0$ is given by $E_b$ since there is only one pair potential defining each dimer.

We then integrate this quadratic form over all of space. The eigenvalues of the Hessian are all $w_i^2 = 2E_0\alpha^2$, yielding

$$\frac{Z_{\text{dimer}}}{Z_1 Z_2} \approx \frac{e^{\beta E_0} \pi^{3/2}}{(\beta E_0)^{3/2} \alpha^3 V} \qquad \text{(Laplace's approximation)}. \qquad (39)$$

While the error from Laplace's approximation in $U_{\text{dimer}}(\vec{q_i})$ itself can be quite significant (at ~40% even for $\beta E_0 = 30$), the error in the resulting yield estimate is much less severe, as seen in Supplementary Fig. 2. The error in both the energy function and the yield go down significantly as more terms are taken in the expansion. The fourth order result shown in Supplementary Fig. 2 is a result of numerical integration of

$$U_{\text{dimer}}(\vec{q_i}) = E_0\left(-1 + \alpha^2 q_i^2 - \alpha^3 q_i^3 + \frac{7177}{12288}\alpha^4 q_i^4\right) + \mathcal{O}(q_i^5). \qquad (40)$$

**The error due to Laplace's approximation decreases for shorter interaction ranges**

As seen in Supplementary Fig. 2, taking Laplace's approximation to second order appears to shift the yield curve to the right compared to the exact calculation. This effect can be quantified for different system conditions by measuring the value of $E_0$ leading to a dimer yield of 1/2. We call this value $E_{1/2}$. In Supplementary Fig. 3 we examine the error in the estimate of $E_{1/2}$ using Laplace's approximation to 2nd order as a function of the interaction range of the spheres $(8/\alpha)$ normalized by the building block diameter $d$. Experimentally, the interaction range is often tunable, for example by changing ionic conditions for protein self-assembly. We find that as expected, the error increases with the interaction range, and especially grows rapidly for interaction ranges larger than the diameter of the spheres.

To perform this comparison we used the same system conditions as in Supplementary Fig. 2, except that in order to ensure that the interaction range remains significantly smaller than the system size, we increased the volume of the system by two orders of magnitude to $1.8 \times 10^6 d^3$. When not performing this modification to the system conditions, a qualitatively similar curve is found, except that the error shows a dramatic spike for large interaction ranges, especially as the range approaches the system size. The error for small interaction ranges is largely unchanged; for example, for normalized interaction ranges ~$10^{-2}$ it is increased by <10% compared to that shown in Supplementary Fig. 3.

### The grand canonical ensemble error for finite systems

While we have used the canonical ensemble to compare our results directly to simulations, most experimental systems are best modeled using the grand canonical (GC) ensemble. What errors are introduced by applying the GC ensemble to finite systems? We consider the spherically symmetric dimer system (Supplementary Fig. 2A) as a case study.

Equation (11) requires us to solve for the ratio of the dimer partition function to the two monomer partition functions. This result was already found in Eq. (37). We denote $4\pi \int r^2 e^{-\beta U(r)} dr$ by $V_{\text{int}}$ for clarity, and let $c_b^{\text{tot}}$ be the total concentration of black monomers (and similarly for white monomers). We let $q$ denote the combination of factors $V_{\text{int}}(c_b^{\text{tot}} + c_w^{\text{tot}}) + 1$. We find that the self-consistent solution to Eqs. (10) and (11) leads to a yield of dimers given by

$$Y_d^{GC} = \frac{c_d}{c_b + c_w + c_d}$$
$$= \frac{q - \sqrt{q^2 - 4V_{\text{int}}^2 c_b^{\text{tot}} c_w^{\text{tot}}}}{q - 2 + \sqrt{q^2 - 4V_{\text{int}}^2 c_b^{\text{tot}} c_w^{\text{tot}}}}. \tag{41}$$

where $c_d$ is the concentration of dimers, and $c_b$ and $c_w$ are the concentrations of free black and white monomers, respectively.

In Supplementary Fig. 5, we compare the result of this self-consistent solution to the result of the exact canonical ensemble calculation considered previously. We place $N_m$ monomers of each kind, with an attractive potential $E_0 = 16$ (in units of $k_B T$) and with an interaction range of $8/\alpha = 8d/5$. We vary $N_m$ and consider two cases for the volume of the system: either a constant volume (of $18,000d^3$ as previously; panel a) or a constant building block density (of $10^{-3}d^{-3}$ as previously; panel b). In both cases, we find that the result of the GC calculation approaches the exact canonical yield as the number of building blocks grows, and does so with a power-law scaling, in agreement with previously published results[30]. While in the grand canonical ensemble, the volume of the system only enters through the concentrations (e.g., $c_b^{\text{tot}} = N_m/V$ in Eq. (41)) the canonical ensemble yield changes as a function of system volume for constant densities, and approaches the grand canonical ensemble prediction in the thermodynamic limit.

### Statistics and reproducibility

No statistical method was used to predetermine sample size. No data were excluded from the analyses. The experiments were not randomized. The Investigators were not blinded to allocation during experiments and outcome assessment.

### Reporting summary

Further information on research design is available in the Nature Portfolio Reporting Summary linked to this article.

## Data availability

Source data are provided with this paper[23], with a highly permissible Apache 2.0 license. See https://doi.org/10.5281/zenodo.8355118.

## Code availability

The code used to carry out this study have been deposited in the github repository[23], with a highly permissible Apache 2.0 license. We have written the code so that the data underlying the Figs. 2–5 can be reproduced by running the underlying code. See https://doi.org/10.5281/zenodo.8355118.

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

## Acknowledgements

We thank Lucy Colwell for suggesting that we use covariance based methods to predict contacts and Yang Hsia, Scott Boyken, Zibo Chen, and David Baker for collaborations on designed protein complexes. We also thank Ned Wingreen for suggesting the alternative derivation of (11). This research was supported by the Office of Naval Research through ONR N00014-17-1-3029, the Simons Foundation the NSF-Simons Center for Mathematical and Statistical Analysis of Biology at Harvard (award number #1764269), the Peter B. Lewis '55 Lewis-Sigler Institute/Genomics Fund through the Lewis-Sigler Institute of Integrative Genomics at Princeton University, and the National Science Foundation through the Center for the Physics of Biological Function (PHY-1734030).

## Author contributions

A.I.C. and M.P.B. conceived of this study. A.I.C., O.K., R.K.K., C.P.G., and M.P.B. contributed technical ideas. Code was written by A.I.C., O.K., R.K.K., and C.P.G. The paper was written by A.I.C., O.K., R.K.K., and M.P.B.

## Competing interests

The authors declare no competing interests.
