## [Peer Review File · Nature Communications]

REVIEWER COMMENTS

Reviewer #1 (Remarks to the Author):

The manuscript reports the development and testing of a novel statistical thermodynamics method to predict the equilibrium yield of complexes composed of identical or non-identical monomers. Such a method is very much needed, as simulations of ensemble processes are inherently time-demanding. The manuscript reads very well, the conclusions are supported by the results presented in the main text and supporting information. My only minor recommendation to the Authors is to include in the revised text a short discussion if and how the presented model could be extended so that it would be valid also for oligomers with zero modes.

Reviewer #2 (Remarks to the Author):

The manuscript by Curatolo et al describes a computational toolbox for calculating the equilibrium assembly yield of heterogeneous building blocks with complex geometries. This is an important problem with relevance to a variety of systems from colloids to protein assembly. The authors argue that simulations are prohibitively expensive for large heterogeneous structures, and that analytical calculations are challenging beyond spherical particles with isotropic interactions. To address this challenge, they use a new computer-based “automatic differentiation” method to simplify and expedite the calculation. Using this approach, they were able to calculate the rotational and translational entropic factors in the free energy of heterogeneous systems and estimate the equilibrium assembly yield. They demonstrated their approach with protein assemblies and multimeric cage structures.

As the authors state, the toolbox may indeed be an important method for a variety of equilibrium problems, and serve as a starting point to deal with complex biological assembly, which typically occurs out of equilibrium.

I am not an expert in the field, yet the calculations seem to me sound and convincing, with the text sufficiently clear. My concern is the accessibility of the manuscript to the general readership. I suspect that only readers with a strong background in theoretical physics and applied math will be able to follow the lengthy technical theoretical calculations and considerations. This might limit the relevance of the present manuscript to the general readership of Nature Comm, for which the main results, intuition and implications are probably best presented in a short communication format.

Reviewer #3 (Remarks to the Author):

The authors use automatic differentiation to streamline standard analytic approaches to the evaluation of cluster partition functions. They apply this method to simulate the self-assembly properties of nanostructures and complex proteins. The use of automatic differentiation in the evaluation of Hessians and Jacobians is not new, and it was in fact a central problem for which these tools were developed by the applied mathematics and computer science community. However, this work appears to be its first application to the problem of self-assembly of clusters. This method appears to provide a large improvement over approaches based on numerical simulation (usually via Monte Carlo sampling), but the exact scale of improvement remains unclear

Overall the work is well motivated and presented, but I have a few major issues with the manuscript.

1) The core of the calculation is in evaluation of the eigenvalues of the Hessian at the ground-state configuration of the cluster, defined as $\xi_i = 0$. However, it is not stated how the ground-state configuration of each cluster was obtained. Did the optimization of the ground state also take advantage of automatic differentiation? Is only the configuration with a global energy minimum considered? Or are some meta-stable configurations with a local energy minima also included? In particular, I am curious how difficult it was to find the ground-state configuration of the 30-sphere cluster in the final example.

2) The link to the github repository (Ref. 20) is garbled with Ref. 21. Can the authors also check if the repository is public (I was unable to find it)? As this work is aimed at providing a new tool for the community, providing a proper open-source repository of the research code is a prerequisite for its publication.

3) Some sort of comparison of this technique's accuracy or speed compared to the standard numerical sampling methods mentioned in the introduction should be performed, to validate the author's claims that the automatic differentiation approach is indeed better. The kind of questions I still have are: How complicated does a problem have to be before the standard numerical sampling approach becomes computational intractable? Does the automatic differentiation approach still work at that scale? Is using automatic differentiation always faster, or only in a certain range of problem complexity?

If these omissions are corrected, I believe the work is suitable for publication in Nature Communications. I also have a few additional minor comments:

4) In the last paragraph on Pg. 5 and the associated Fig. 3, is "self assembly" defined as a yield > 50%? Also, this section should probably state that $D_0 \sim 20$ if $\text{Beta } E_0 = 50$ (for easier comparison to the figure).

5) How was the "simulation" for Fig. S2 performed? What method or code was used?

6) There appears to be an erroneous "red" LaTeX command appearing throughout the manuscript, making some interpretation difficult (e.g. pg 2, just before Eq. 3). Also, some of the Equation references have double parentheses, e.g. ((3)) instead of (3), especially in the supplementary material.

Reviewer #4 (Remarks to the Author):

In "The assembly yield of complex, heterogeneous structures: A computational toolbox," Curatolo et al. develop a combined analytical/computational method for computing the partition function of arbitrarily complex and heterogeneous building blocks that self-assemble into rigid clusters. Via the calculation of the partition function, yield of the clusters can be predicted under continuously varying system parameters. The authors then use this method in several example test cases, including patchy particle formation into rigid dimers, protein complex self-assembly for two different protein complexes, and spherical cage formation. The authors find that their method, which is enabled by automatic differentiation, is capable of predicting yield in these separate cases, demonstrating its potential power and use as a tool for understanding yield behavior (as well as calculating other thermodynamic quantities that depend on the partition function) in highly complex systems.

The authors' combined analytical/computational approach represents an important step forward for the field and opens innumerable doors to more complex problems. I believe that the results of this study will be of interest to theoretical, computational, and experimental communities, and that this work is suitable for publication in Nature Communications, with some additional clarification of the methods.

I have the following suggestions/questions for the authors:

(1) In general, as far as I understand, the power and novelty of your approach in part lies on the incorporation of automatic differentiation. I suggest including some broad introduction of what exactly this process entails, since your readers might be quite unfamiliar with it.

(2) Also, since this is a paper primarily introducing a method, I think it would be useful to provide more details regarding its efficiency or the computational cost required to employ it in the contexts you considered. I found myself wondering exactly how long/how many resources it took to sample Eqn (8), and how that scaled with system size and cluster geometry. Can you provide any intuition along those lines? Why did you choose to sample 100,000 cluster orientations, rather than any other number? Did you find that this sample size was the minimum required for some convergence in the integral, or were you limited by computational cost?

(3) The authors mention that the error between the theoretical curve and the simulations, in their first case study, arises from Laplace's approximation in calculating the partition function. I very much appreciated the section in the SI exploring this further. I think it would be useful in the main document to include a little more discussion of this error, as a helpful guide as others might begin to apply this method to their own systems. For example, do the authors have more general intuition for under what conditions this error would be smaller or larger? Is the result in Fig. S3 general for any interaction? Does this error increase with system size or is it independent of that?

(4) This method requires that the clusters be rigid. Could it be extended to calculate the partition function for floppy structures, or is that completely analytically intractable?

(5) Perhaps I have misunderstood, but once you have the partition function of the system, and are able to automatically differentiate it, have you essentially unlocked all the thermodynamic variables related to the system? You do not mention this in the paper, but I find the prospect of calculating these properties beyond yield very exciting- perhaps the authors could mention the possibilities related to this point somewhere in the manuscript. For example, is it possible to calculate free energy barriers associated with nucleation and growth for systems of arbitrarily complex building blocks using this method?

My remaining suggestions are more minor:

(6) The word "red" appeared several times throughout the manuscript where it was not supposed to appear, I believe.

(7) On page 2, in the last paragraph, you mention that rigid clusters are clusters "without zero modes." I might be more specific about what you mean here, for the readers who are less familiar with deformation modes in mechanical systems- perhaps mention which modes you are referring to and why they are "zero."

(8) In Eqn 10, do you account for the free monomers? Is that one of the structures enumerated by "s"?

(9) In your discussion of overall energy of the TRAP complex you introduce 6 new "D" parameters (D_X, D_MO, etc) without ever defining them. I found myself very confused here, trying to keep track of all of these parameters, and not exactly knowing what each represented. Could the authors provide more explanation in the text to this effect?

Reviewer #1

To clarify our responses, we put sections reprinted from the manuscript in **red**, while new discussions added to the manuscript are in **blue**. Reviewer comments are in *italics*.

Reviewer Comments

The manuscript reports the development and testing of a novel statistical thermodynamics method to predict the equilibrium yield of complexes composed of identical or non-identical monomers. Such a method is very much needed, as simulations of ensemble processes are inherently time-demanding. The manuscript reads very well, the conclusions are supported by the results presented in the main text and supporting information. My only minor recommendation to the Authors is to include in the revised text a short discussion if and how the presented model could be extended so that it would be valid also for oligomers with zero modes.

We thank the reviewer for this suggestion and agree that a short discussion of how the model could be extended to zero modes is warranted. We have revised the relevant section as follows:

We restrict our calculation to only treat rigid clusters, where we use the term “rigid” as opposed to “floppy” to indicate a cluster without zero modes (i.e. internal degrees of freedom about which movement incurs no energetic cost) [26]. Accounting for such internal degrees of freedom requires a challenging and nontrivial calculation of the relevant entropic factor beyond the scope of this current work. A geometric formulation of the entropic factor resulting from such floppy modes has been addressed for the case of isotropic potentials with short interaction range in Ref. [27]. Intuitively, the free energy of a system with n zero modes is represented as an n -dimensional manifold whose boundaries can be determined from the collection of configurations with $(n-1)$ zero modes.

Reviewer #2

To clarify our responses, we put sections reprinted from the manuscript in **red**, while new discussions added to the manuscript are in **blue**. Reviewer comments are in *italics*.

Reviewer Comments

The manuscript by Curatolo et al describes a computational toolbox for calculating the equilibrium assembly yield of heterogeneous building blocks with complex geometries. This is an important problem with relevance to a variety of systems from colloids to protein assembly. The authors argue that simulations are prohibitively expensive for large heterogeneous structures, and that analytical calculations are challenging beyond spherical particles with isotropic interactions. To address this challenge, they use a new computer-based “automatic differentiation” method to simplify and expedite the calculation. Using this approach, they were able to calculate the rotational and translational entropic factors in the free energy of heterogeneous systems and estimate the equilibrium assembly yield. They demonstrated their approach with protein assemblies and multimeric cage structures.

As the authors state, the toolbox may indeed be an important method for a variety of equilibrium problems, and serve as a starting point to deal with complex biological assembly, which typically occurs out of equilibrium.

In am not an expert in the field, yet the calculations seem to me sound and convincing, with the text sufficiently clear. My concern is the accessibility of the manuscript to the general readership. I suspect that only readers with a strong background in theoretical physics and applied math will be able to follow the lengthy technical theoretical calculations and considerations. This might limit the relevance of the present manuscript to the general readership of Nature Comm, for which the main results, intuition and implications are probably best presented in a short communication format.

We thank the reviewer for their general comments, particularly their expressed concern regarding the accessibility of our manuscript. To address this concern, we have rewritten the “Conclusion” section of our manuscript to better communicate the main results, intuition, and implications to a general audience. For clarity, we have copied this revised section below. Specifically, note the references to Equations 9, 10, and 11 in the second paragraph – we consider this a “user guide” to our work:

Many structural and functional properties of biological systems rely on heterogeneous self assembly. Increasingly, building blocks with highly-tuned geometries and interactions can be made experimentally, presenting an opportunity to design complex self-assembling systems with properties of living matter [46-48]. However, for increasing numbers of individual components, the yield of the desired product is diluted by an exponential number of off-target assemblies. Even if each of the off-target assemblies forms with a lower probability than the desired product, the multiplicity of these off-target assemblies can cause a yield catastrophe in which the desired

product forms with negligible yield [49]. An efficient method for predicting assembly yield would enable practitioners to design self-assembling systems of increasing complexity while mitigating this yield catastrophe.

In this work, we have developed a combined analytical/computational approach to calculate the equilibrium assembly yields of complexes comprised of heterogeneous building blocks with arbitrary geometries. Our approach involves two novel calculations: (i) computing the partition function of each (predefined) candidate complex (see Equation 9) and (ii) given these partition functions, calculating equilibrium assembly yields (see Equations 10 and 11). The classical statistical mechanics methodology we put forth can be realized by modern automatic differentiation techniques enabled by advances in machine learning [21]. While previously, the efficient calculation of the entropic parts of the partition function has been intractable except for simple cases, these advances make this calculation possible even for structures with complicated building blocks or with a large number of different components.

Our theoretical framework provides significant conceptual and practical improvements over simulation-based methods, the only other general-purpose method for yield prediction. Calculations via our approach require a drastically reduced computational cost compared to simulations (i.e. seconds or minutes vs. hours, weeks, or months). Moreover, simulation-based methods can be fraught with additional difficulties such as issues in simulating finite concentrations or efficiently sampling from equilibrium distributions [50]. Lastly, since our method involves the direct calculation of the partition function, our approach can also be used to compute other statistical properties of an equilibrium thermodynamic system (e.g. heat capacity, energy fluctuations) without complicated modifications to the calculation.

We have demonstrated the validity of this model's approximations by comparison to molecular dynamics simulations, where we found that precise agreement between the simulations and the analytical calculations requires paying close attention to higher order corrections to the Laplace approximations. We have also applied our methodology to two illustrative protein complexes (PFL and TRAP), as well as a simple example of a cage-forming system. We emphasize that our method can be applied to arbitrary proteins complexes so long as we know how to construct the building blocks, calculate the energies of contacts, and enumerate the most likely structures. The latter is in principle a difficult problem: enumerating all possible combinations of building blocks can become infeasible for complexes comprised of large numbers of building blocks.

We next plan to apply this method to self-assembling systems that are the subject of active study, such as *de novo* designed proteins and viral capsids [51]. In fact, when designing *de novo* protein complexes, significant care must be taken to mitigate the self-assembly of off-target structures [49]. To address this problem *in silico*, our model can be combined with state-of-the-art protein structure prediction and a suite of existing energy functions for describing protein-protein interactions [52-54].

Finally, as we have described, we have restricted our calculations to equilibrium systems. However, biological systems frequently make use of non-equilibrium control elements, including

the regulation of the production machinery for the assembled components, as well as other control elements that are not yet available in synthetic systems such as allosteric interactions. These control knobs provide more ways of regulating energetic and entropic interactions leading to self assembly. Future work may seek to provide an understanding of how these control elements contribute to high self-assembly yields, and thus test which would be most effective to prioritize developing in a synthetic context.

Reviewer #3

To clarify our responses, we put sections reprinted from the manuscript in red, while new discussions added to the manuscript are in blue. Reviewer comments are in *italics*.

Summary

The authors use automatic differentiation to streamline standard analytic approaches to the evaluation of cluster partition functions. They apply this method to simulate the self-assembly properties of nano-structures and complex proteins. The use of automatic differentiation in the evaluation of Hessians and Jacobians is not new, and it was in fact a central problem for which these tools were developed by the applied mathematics and computer science community. However, this work appears to be its first application to the problem of self-assembly of clusters. This method appears to provide a large improvement over approaches based on numerical simulation (usually via Monte Carlo sampling), but the exact scale of improvement remains unclear

Specific Comments

Overall the work is well motivated and presented, but I have a few major issues with the manuscript.

1) The core of the calculation is in evaluation of the eigenvalues of the Hessian at the ground-state configuration of the cluster, defined as $x_i = 0$. However, it is not stated how the ground-state configuration of each cluster was obtained. Did the optimization of the ground state also take advantage of automatic differentiation? Is only the configuration with a global energy minimum considered? Or are some meta-stable configurations with a local energy minima also included? In particular, I am curious how difficult it was to find the ground-state configuration of the 30-sphere cluster in the final example.

We thank the reviewer for their helpful feedback. Regarding how the ground states were obtained for the assemblies considered in the presented experiments, the method varied by the type of assembly. For the protein examples in Figures 3 and 4, we considered the structure given in the corresponding PDB file as the ground state. More specifically, we define a pair of attractive patches at the midpoint of each bond such that their initialized positions minimize the potential energy; for this reason, no minimization (e.g. via simulation) is required. We explain this procedure in the section titled "Predicting protein complex yield" and have added a sentence to clarify the lack of need for minimization:

We begin by using the PDB file to define coarse-grained, rigid building blocks: each amino acid in the protein is represented by a sphere whose position is a non-weighted average of the positions of the amino acid's atoms. Similarly to Fig. 2A, the interface contacts are defined by patches placed on the interface (see Fig. 1), subjected to an attractive pair potential (see Supplementary section D). For example, if it is known that amino acid *a* belonging to protein A forms a bond with amino acid *b* belonging to protein B and their positions in the complex are (x_a, y_a, z_a) and (x_b, y_b, z_b) respectively, then a patch on each protein building block will be set at

position $(x_a+x_b, y_a+y_b, z_a+z_b)/2$; these two patches specifically attract one another. The interaction range is set at $8/\alpha=8d/2$ where d is the diameter of each sphere representing an amino acid. Since the patches are defined so as to minimize energy, no additional minimization procedure (e.g. simulation) is required to define the ground state.

In the case of the sphere-packing experiments, we used a previously reported set of coordinates for the ground states (see Reference 45). We refer to this in the section titled “The yield of spherical cages” but have added an additional sentence for clarity:

We present a notional example of such an assembly, in an effort to demonstrate how this formalism works. We consider a cage of radius $R=1$ that is assembled out of 60 spherical building blocks with icosahedral symmetry. We use a set of previously reported coordinates to define the ground state of each assembly [45].

2) The link to the github repository (Ref. 20) is garbled with Ref. 21. Can the authors also check if the repository is public (I was unable to find it)? As this work is aimed at providing a new tool for the community, providing a proper open-source repository of the research code is a prerequisite for its publication.

We have cleaned the code associated with our experiments and made it publicly available at the following GitHub repository: <https://github.com/rkruegs123/self-assembly-toolkit>. We have also modified the citation to the GitHub repository accordingly.

3) Some sort of comparison of this technique's accuracy or speed compared to the standard numerical sampling methods mentioned in the introduction should be performed, to validate the author's claims that the automatic differentiation approach is indeed better. The kind of questions I still have are: How complicated does a problem have to be before the standard numerical sampling approach becomes computational intractable? Does the automatic differentiation approach still work at that scale? Is using automatic differentiation always faster, or only in a certain range of problem complexity?

We agree with the reviewer that a more explicit comparison to existing approaches would better place our theory in context. Given the challenges of explicit simulations—namely speed and convergence issues, especially when simulating large numbers of particles—we believe that in those cases in which our theory is applicable (i.e. finite concentrations, explicit enumeration of possible structures, no floppy modes) our theory would be preferable to simulations, especially in terms of speed. In fact, our theory works best in those cases that are most challenging for simulations: when considering extremely large numbers of particles (see Fig. S5). In order to further compare our theory to current state-of-the-art approaches (beyond Figs. 2, S2, and S3 which compare our theory to simulations), we have revised Figure 2 to include calculations of the dimer yield in which entropic contributions to the partition function are ignored. These simpler (though currently state-of-the-art) calculations appear to be highly inaccurate, while the

theory developed in this work closely corresponds to the yield values calculated via simulation. We have revised the text to describe this change as follows:

In order to compare the analytical theory to simulation most directly, we operate in the canonical ensemble, where we can perform exact calculations corresponding to finite sized simulations (see Supplementary section E). The comparison between simulations and theory is shown in Fig. 2. We also show the predictions arising from neglecting the contributions of entropy to the partition function, with $Z_d/Z_m = e^{3\beta E_b}$. The result of $Y_d = Z_d/(Z_d + Z_m)$ is shown in brown, while a prediction accounting for finite concentrations using Eqns. 10 & 11 is shown in orange.

The revised Figure 2 is copied below:

If these omissions are corrected, I believe the work is suitable for publication in Nature Communications. I also have a few additional minor comments:

4) In the last paragraph on Pg. 5 and the associated Fig. 3, is "self assembly" defined as a yield > 50%? Also, this section should probably state that $D_0 \sim 20$ if $\text{Beta } E_0 = 50$ (for easier comparison to the figure).

We agree with the reviewer that "self assembly" is not defined clearly in this section. We have modified the text to clarify that "self assembly" is indeed defined as yield > 50%. We also agree

with the reviewer regarding the change in text for easier comparison to the figure and have made the corresponding change to the text:

For example, for $\epsilon \approx 20$ kBT (corresponding to energies of individual contacts ranging from $E_{AB}^{(b)} = 6$ kBT to $E_{AB}^{(b)} = 20$ kBT) the model predicts self-assembly (i.e. dimer yield > 50%) for concentrations $\geq 10^{-5} \text{d}^{-3}$ (~16.6 mM).

5) How was the "simulation" for Fig. S2 performed? What method or code was used?

All simulations were performed using HOOMD-blue. We added a brief sentence describing this on page 5, in the description of Fig. 2 (the first time a simulation we perform is referenced):

All simulations were performed in the HOOMD-blue simulation package [33].

6) There appears to be an erroneous "red" LaTeX command appearing throughout the manuscript, making some interpretation difficult (e.g. pg 2, just before Eq. 3). Also, some of the Equation references have double parentheses, e.g. ((3)) instead of (3), especially in the supplementary material.

We thank the reviewer for pointing out these formatting errors. The erroneous "red" commands were an artifact of differences in LaTeX compilers – we have removed all instances of this. Similarly, we have removed all instances of double parentheses.

Reviewer #4

To clarify our responses, we put sections reprinted from the manuscript in red, while new discussions added to the manuscript are in blue. Reviewer comments are in *italics*.

Summary

In “The assembly yield of complex, heterogeneous structures: A computational toolbox,” Curatolo et al. develop a combined analytical/computational method for computing the partition function of arbitrarily complex and heterogeneous building blocks that self-assemble into rigid clusters. Via the calculation of the partition function, yield of the clusters can be predicted under continuously varying system parameters. The authors then use this method in several example test cases, including patchy particle formation into rigid dimers, protein complex self-assembly for two different protein complexes, and spherical cage formation. The authors find that their method, which is enabled by automatic differentiation, is capable of predicting yield in these separate cases, demonstrating its potential power and use as a tool for understanding yield behavior (as well as calculating other thermodynamic quantities that depend on the partition function) in highly complex systems.

The authors’ combined analytical/computational approach represents an important step forward for the field and opens innumerable doors to more complex problems. I believe that the results of this study will be of interest to theoretical, computational, and experimental communities, and that this work is suitable for publication in Nature Communications, with some additional clarification of the methods.

Specific Comments

I have the following suggestions/questions for the authors:

(1) In general, as far as I understand, the power and novelty of your approach in part lies on the incorporation of automatic differentiation. I suggest including some broad introduction of what exactly this process entails, since your readers might be quite unfamiliar with it.

We thank the reviewer for their helpful feedback and agree that a more thorough introduction to automatic differentiation is warranted. We have added the following text to the introduction to accomplish this:

Another approach to predict the equilibrium self-assembly yield is to calculate it analytically. While for spherical particles with isotropic interactions the partition functions for small clusters can be calculated analytically [17], no such analytical calculation exists for anisotropic interactions. We hypothesized that automatic differentiation could be leveraged to perform this otherwise intractable calculation [18-20]. In automatic differentiation, the execution of a computer program is accompanied by the construction of a computation graph of primitive operations whose derivatives are known and can therefore be recombined (via the chain rule) to compute the gradient of the larger program. This procedure can be applied recursively, allowing us to efficiently evaluate higher-order derivatives of nearly any computer function with machine accuracy.

(2) Also, since this is a paper primarily introducing a method, I think it would be useful to provide more details regarding its efficiency or the computational cost required to employ it in the contexts you considered. I found myself wondering exactly how long/how many resources it took to sample Eqn (8), and how that scaled with system size and cluster geometry. Can you provide any intuition along those lines? Why did you choose to sample 100,000 cluster orientations, rather than any other number? Did you find that this sample size was the minimum required for some convergence in the integral, or were you limited by computational cost?

Since the calculation of the rotational entropy via sampling 100,000 cluster orientations is the primary computational cost of our method, we added a sentence describing the computational cost of this sampling procedure. It may be possible to achieve high accuracy with significantly fewer calculations but this was not necessary for our purposes, and we found that 100,000 was sufficiently large to yield accurate predictions.

First, we uniformly sample 10^5 values of φ_c using the quaternion representation, to avoid problems that arise from sampling Euler angles, such as non-uniform distribution of orientations, singularities, and the gimbal lock problem [20]. It may be possible to achieve high accuracy with fewer calculations but we found that 10^5 samplings were sufficient to yield accurate analytic predictions. For the results presented in this manuscript, this sampling procedure took <30 seconds of compute time and <3 MB of memory (as measured by the `tracemalloc` library) on a personal laptop computer (e.g. 26s and 2 MB for the TRAP complex trimer presented in Figure 4).

Note that since the particles comprising the spherical cages interact isotropically, the calculation of the rotational entropy could be computed analytically. We have also added the following sentence to clarify this, in our description of the spherical cage analysis:

Because the spheres interact with an isotropic potential, it was not necessary to use Equation 8 to calculate the rotational partition function, and we instead calculate this rotational component from the moments of inertia [17].

(3) The authors mention that the error between the theoretical curve and the simulations, in their first case study, arises from Laplace's approximation in calculating the partition function. I very much appreciated the section in the SI exploring this further. I think it would be useful in the main document to include a little more discussion of this error, as a helpful guide as others might begin to apply this method to their own systems. For example, do the authors have more general intuition for under what conditions this error would be smaller or larger? Is the result in Fig. S3 general for any interaction? Does this error increase with system size or is it independent of that?

We have revised the paragraph where we introduce our use of Laplace's approximation to include a discussion of the error that the approximation introduces:

Here, we have taken Laplace's approximation to lowest 2nd order. The error arising from this approximation will increase for less parabolic energy landscapes such as can occur for more esoteric interaction potentials, more complex interfaces, or even for longer-range interactions (Fig. S3). As energy minima become less parabolic, higher-order corrections need also be considered. Such corrections can be computed using the inverse of the Hessian matrix alongside higher-order derivatives [28]. In the Numerical Results section we explore and comment on the validity of Laplace's approximation in this context.

(4) This method requires that the clusters be rigid. Could it be extended to calculate the partition function for floppy structures, or is that completely analytically intractable?

Extending the model to calculate the partition for "floppy structures" is not necessarily analytically intractable, but is nontrivial and beyond the scope of our work. We have revised the relevant section as follows:

We restrict our calculation to only treat rigid clusters, where we use the term "rigid" as opposed to "floppy" to indicate a cluster without zero modes (i.e. internal degrees of freedom about which movement incurs no energetic cost) [26]. Accounting for such internal degrees of freedom requires a challenging and nontrivial calculation of the relevant entropic factor beyond the scope of this current work. A geometric formulation of the entropic factor resulting from such floppy modes has been addressed for the case of isotropic potentials with short interaction range in Ref. [27]. Intuitively, the free energy of a system with n zero modes is represented as an n -dimensional manifold whose boundaries can be determined from the collection of configurations with $(n-1)$ zero modes.

(5) Perhaps I have misunderstood, but once you have the partition function of the system, and are able to automatically differentiate it, have you essentially unlocked all the thermodynamic variables related to the system? You do not mention this in the paper, but I find the prospect of calculating these properties beyond yield very exciting- perhaps the authors could mention the possibilities related to this point somewhere in the manuscript. For example, is it possible to calculate free energy barriers associated with nucleation and growth for systems of arbitrarily complex building blocks using this method?

We agree that since we calculate the partition function of the system, our method can be used to study all related thermodynamic variables. One caveat to using our method to studying processes like nucleation and growth is that the intermediate states corresponding to free energy barriers must be known. Interestingly, our method could also be adapted for inverse design as recently developed algorithms permit the efficient computation of gradients with respect to numerical optimization procedures via implicit differentiation (e.g. see

https://jaxopt.github.io/stable/implicit_diff.html). We have added the following sentences to the conclusion to highlight these properties of our model:

Moreover, simulation-based methods can be fraught with additional difficulties such as issues in simulating finite concentrations or efficiently sampling from equilibrium distributions [50]. Lastly, since our method involves the direct calculation of the partition function, our approach can also be used to compute other statistical properties of an equilibrium thermodynamic system (e.g. heat capacity, energy fluctuations) without complicated modifications to the calculation. Indeed, since the gradient of a solution to Equations 10 and 11 can be computed implicitly [51], our method could enable inverse design with respect to related thermodynamic equilibrium properties.

My remaining suggestions are more minor:

(6) The word “red” appeared several times throughout the manuscript where it was not supposed to appear, I believe.

We thank the reviewer for pointing out these formatting errors. The erroneous “red” commands were an artifact of differences in LaTeX compilers – we have removed all instances of this.

(7) On page 2, in the last paragraph, you mention that rigid clusters are clusters “without zero modes.” I might be more specific about what you mean here, for the readers who are less familiar with deformation modes in mechanical systems- perhaps mention which modes you are referring to and why they are “zero.”

To make our discussion of rigidity accessible to a wider audience, we have added the following clarification in our discussion of zero modes:

We restrict our calculation to only treat rigid clusters, where we use the term “rigid” as opposed to “floppy” to indicate a cluster without zero modes (i.e. internal degrees of freedom about which movement incurs no energetic cost) [26].

(8) In Eqn 10, do you account for the free monomers? Is that one of the structures enumerated by “s”?

Yes. When introducing Eqn 10, we have added the following sentence to clarify this point:

We therefore seek a self-consistent solution for the yields of the different structures, while imposing conservation laws for each of the monomer species. There is one conservation law for each monomer species, given by... and as previously $N_{s, \alpha}$ is the number of monomers of type α in each structure s . Note that the unbound monomer is a valid equilibrium structure and is therefore included in the above sum.

(9) In your discussion of overall energy of the TRAP complex you introduce 6 new “D” parameters (D_X, D_MO, etc) without ever defining them. I found myself very confused here, trying to keep track of all of these parameters, and not exactly knowing what each represented. Could the authors provide more explanation in the text to this effect?

We thank the reviewer for their suggestion – we agree that the presentation of these parameters was relatively unclear in our original submission. We have completely revised the notation used in this section for clarification. For brevity, we refer the reviewer to the section titled “Predicting protein complex yield” and the corresponding Figures 3 and 4.

REVIEWERS' COMMENTS

Reviewer #3 (Remarks to the Author):

The authors have sufficiently responded to my concerns and I find the paper suitable for publication.

Reviewer #4 (Remarks to the Author):

I very much appreciate the authors' efforts to clarify their methods throughout the manuscript, and I especially appreciate the additional detail regarding the protein complex simulations. I recommend the manuscript for publication, and have no further suggestions or comments. I look forward to reading about the authors' future work in this arena!